



# A Comparative Evaluation of Aura-OMI and SKYNET Near-UV Single-scattering Albedo Products

Hiren Jethva[1,2]*, Omar Torres[2]

[1]Universities Space Research Association, Columbia, MD 21044 USA

[2]NASA Goddard Space Flight Center, Greenbelt, MD 20771 USA

**Mailing Address:**

Room#A422, Building#33,

Laboratory of Atmospheric Chemistry & Dynamics

Earth Science Division

NASA Goddard Space Flight Center,

Greenbelt, MD 20771, USA

*Corresponding Author: Dr. Hiren Jethva

E-mail: hiren.t.jethva@nasa.gov



**ABSTRACT**
The aerosol single-scattering albedo (SSA) retrieved by the near-UV algorithm applied to the
Aura/Ozone Monitoring Instrument (OMI) measurements (OMAERUV) is compared with an
independent inversion product derived from the sky radiometer network SKYNET-a ground-
based radiation observation network span over Asia and Europe. The present work continues
our efforts to evaluate the consistency between the retrieved SSA from satellite and ground
sensors. The automated spectral measurements of direct downwelling solar flux and sky
radiances made by SKYNET Sun-sky radiometer are used as input to an inversion algorithm that
derives spectral aerosol optical depth (AOD) and single-scattering albedo (SSA) in the near-UV
to near-IR spectral range. The availability of SKYNET SSA measurements in the ultraviolet region
of the spectrum allows, for the first time, a direct comparison with OMI SSA retrievals
eliminating the need of extrapolating the satellite retrievals to the visible wavelengths as the
case in the evaluation against the Aerosol Robotic Network (AERONET).  An analysis of the
collocated retrievals from over 25 SKYNET sites reveals that about 61% (84%) of OMI-SKYNET
matchups agree within the absolute difference of ±0.03 (±0.05) for carbonaceous aerosols, 50%
(72%) for dust aerosols, 45% (75%) for urban-industrial aerosol types. Regionally, the
agreement between the two inversion products was robust over several sites in Japan
influenced by carbonaceous and urban-industrial aerosols, at the biomass burning site *Phimai*
in Thailand, and polluted urban site in *New Delhi*, India. The collocated dataset yields fewer
matchups identified as dust aerosols mostly over the site *Dunhuang* with more than half of the
matchup points confined to within ±0.03 limits. Combinedly, the OMI-SKYNET retrievals agree
mostly within ±0.03 for the AOD (388 or 400 nm) larger than 0.5 and UV Aerosol Index larger
than 0.2. The remaining uncertainties in both inversion products can be attributed to specific
assumptions made in the retrieval algorithms, i.e., the uncertain calibration constant,
assumption of spectral surface albedo and particle shape, and sub-pixel cloud contamination.
The assumption of fixed and spectrally neutral surface albedo (0.1) in the SKYNET inversion
appears to be unrealistic, leading to a large underestimation of retrieved SSA, especially for low
aerosol load conditions. At large AOD values for carbonaceous and dust aerosols, however,



retrieved SSA values by the two independent inversion methods are generally consistent in
spite of the differences in retrieval approaches.



## 1 INTRODUCTION


Satellite-based remote sensing of aerosols has become an essential tool to detect, quantify, and
routinely monitor the aerosol optical and size properties over the globe. An accurate
representation of aerosols in the climate models is an essential requirement for reducing the
uncertainty in aerosol-related impact on the Earth's radiation balance (direct and semi-direct
effects) and cloud microphysics (indirect effect) (*IPCC*, 2013). The fundamental aerosol
parameters determining the strength and sign of the radiative forcing are the aerosol optical
depth (AOD) and single-scattering albedo (SSA) in addition to the reflective properties of the
underlying surface. While the columnar AOD represents the total extinction (scattering and
absorption) resulting from the interactions with solar radiation, SSA describes the relative
strength of scattering to the total extinction. Together, both AOD and SSA determine the
magnitude and sign of the aerosol radiative forcing. For example, a decrease in SSA from 0.9 to
0.8 can often change the sign of radiative forcing from negative (cooling) to positive (warming)
that also depends on the albedo of the underlying surface and the altitude of the aerosols
(*Hansen et al.*, 1997). Thus, an accurate estimate of both quantities is a prime requirement for
reliable estimates of the net effect of atmospheric aerosols produced with the anthropogenic
as well as natural activities.

Launched in July 2004, the Ozone Monitoring Instrument (OMI) onboard NASA's Aura satellite
has now produced more than a decade long global record of observations of reflected radiation
from Earth in the 270–500 nm wavelength range of the spectrum on a daily basis. OMI scans
the entire Earth in 14 to 15 orbits with its cross-track swath of ~2600 km at ground level at a
nadir ground pixel spatial resolution of $13 \times 24$ km². Satellite observations of the top-of-
atmosphere reflected light at 354 and 388 nm wavelengths made by OMI are used to derive the
UV aerosol index (UVAI) as well as the AOD and SSA using a near-UV algorithm (OMAERUV) that
takes advantage of the well-known sensitivity to the aerosol absorption in the UV spectral
region (*Torres et al.*, 1998). While a general description of the OMI/OMAERUV algorithm is
presented in *Torres et al.* (2007), the recent algorithmic upgrades are documented in *Torres et*





*al.* (2013, 2018). The most important changes applied in the latest OMAERUV algorithm
upgrade includes: 1) use of new carbonaceous aerosol models that account for the presence of
organics in the carbonaceous aerosols by assuming wavelength-dependent imaginary part of
the refractive index (*Jethva and Torres*, 2011), 2) an implementation of robust scheme to
identify aerosol type (smoke, dust, urban/industrial) that combinedly uses the information on
carbon monoxide (CO) observations from the Atmospheric Infrared Sounder (AIRS) and UVAI
from OMI (*Torres et al.*, 2013), 3) use of the aerosol height climatology dataset derived from
the Cloud-Aerosol Lidar with Orthogonal Polarization (CALIOP) lidar-based measurements of
the vertical profiles of aerosol for the carbonaceous and dust aerosols (*Torres et al.*, 2013), and
4) better treatment of dust particles assuming realistic spheroidal shape distribution (*Torres et*
*al.*, 2018). Additionally, the upgraded OMAERUV algorithm has adopted a new method to
calculate UVAI, which now accounts for the angular scattering effects of clouds and significantly
reduces a scan angle related asymmetry in UVAI over cloudy scenes (*Torres et al.*, 2018).

The present work continues our efforts to evaluate the consistency between ground-based SSA
measurements and satellite retrievals from near UV observations. On the first attempt to
intercompare space-based and surface near UV SSA measurements, Earth Probe TOMS
retrievals were compared to AERONET observations acquired during the SAFARI 2000 field
campaign (*Torres et al.*, 2005). The OMAERUV near-UV aerosol product of AOD and SSA has
been continually assessed and validated against the ground-based measurements acquired
from the globally distributed Aerosol Robotic Network-AERONET (*Torres et al.*, 2007; *Ahn et al.*,
2008; *Jethva and Torres*, 2011; *Ahn et al.*, 2014; *Jethva et al.*, 2014). While the OMAERUV AOD
product was directly validated against the AERONET measurements made in the near-UV (340-
380 nm), as carried out in *Ahn et al.* (2014), the SSA retrievals have been evaluated by
comparing with an independent ground  inversion product of AERONET by *Jethva et al.* (2014).
The later analysis required OMI retrievals of SSA to be extrapolated to the shortest visible
wavelength of 440 nm of AERONET inversion product to make the comparison possible. Such
adjustment in the wavelength of retrievals can introduce uncertainty in the comparison arising



from inaccuracy of the spectral dependence of absorption assumed in the wavelength
conversion.

A direct comparison of the column-integrated SSA at 388 nm retrieved from OMI requires
equivalent ground-based columnar retrievals in the near-UV region. The international network
of scanning sun-sky radiometers (SKYNET) fulfills this requirement as it performs the direct Sun
and sky measurements in the near-UV (340-380 nm) as well as visible/near-IR (400-1020 nm)
regions of the spectrum, and derives spectral AOD and SSA at these wavelengths. Taking
advantage of the availability of ground-based SSA inversions in the near-UV from SKYNET,  we
inter-compare the  OMI and SKYNET SSA products at several SKYNET sites in Asia and Europe.
Since both retrieval approaches are based on inversion algorithms that rely on assumptions, the
resulting level of agreement can only be interpreted as a measure of consistency (or lack
thereof) in the measurement of the same physical parameter by fundamentally different
remote sensing approaches.

The paper is organized as follows: section 2 describes the satellite and ground-based data sets
assessed in this analysis along with the collocation methodology; the results of OMI-SKYNET
SSA comparison over individual sites, combinedly for each aerosol type, and diagnosis of
differences between them are presented in section 3;  the possible sources of uncertainty in
both inversion products are discussed in section 4; the paper is summarized and concluded in
section 5.

## 2   DATASETS
### 2.1   THE OMI-OMAERUV AEROSOL PRODUCT
The entire record of OMI observations (October 2004 to present) has been reprocessed
recently with the refined OMAERUV algorithm (PGEVersion V1.8.9.1) to derive a comprehensive





aerosol product that includes the retrievals of UV Aerosol Index (UVAI), AOD, SSA, and AAOD
(388 nm) at a pixel resolution of 13 x 24 km$^2$ at nadir viewing geometry. The retrieved
parameters are also reported at 354 nm and 500 nm wavelengths following the spectral
dependence of aerosols assumed in the chosen model. The data set is available in the HDF-
EOS5 format and can be obtained at no cost from NASA Goddard Earth Sciences (GES)-Data and
Information Services Center (DISC) server at http://daac.gsfc.nasa.gov/. The recent upgrade has
been documented in detail in the work of *Jethva and Torres* (2011), *Torres et al.* (2013, 2018)
and *Ahn et al.* (2014). Here, we use the OMAERUV Level 2 Collection 003 (V1.8.9.1) aerosol
product processed in July 2017.

Post-2007, the OMI observations have been affected by a possible external obstruction that
perturbs both the measured solar flux and Earth radiance. This obstruction affecting the quality
of radiance at all wavelengths for a particular viewing direction is referred to as "row anomaly"
(RA) since the viewing geometry is associated with the row numbers on the charge-coupled
device detectors. The RA issue was detected first time in mid-2007 with a couple of rows which
during the later period of operation expanded to other rows in 2008 and later. At present,
about half of the total 60 rows across the track are identified and flagged as row anomaly
affected positions for which no physical retrievals are performed (*Schenkeveld et al.*, 2017). The
details about this issue can be found at
http://www.knmi.nl/omi/research/product/rowanomaly-background.php. The RA has
significantly affected the sampling during post-2008 OMI measurements, where about half of
the OMI swath is blanketed by row anomaly flags. As a result, the availability of the number of
retrievals over a particular station is reduced starting in 2009 compared to earlier OMI
measurements. Consequently, the OMI-SKYNET matchups are also expected to be lower during
the row anomaly affected period. The OMAERUV algorithm assigns quality flags to each pixel
which carries information on the quality of the retrieval depending upon the observed
condition. We used aerosol retrievals free of RA and flagged as quality flag '0', which are
considered best in accuracy due to higher confidence in detecting aerosols in a scene with
minimal or no cloud contamination.



## 2.2  THE SKYNET AEROSOL INVERSION PRODUCT

The SKYNET is an international network of scanning sun-sky radiometers (manufactured by *Prede Co. Ltd.,* Japan) performing routine and long-term measurements of direct and diffuse solar radiations at several wavelengths spanning UV (340 and 380 nm), visible (400, 500, 675 nm), near-IR region (875, 1020 nm), and in shortware-IR (1627 nm and 2200 nm)  of the spectrum. The automated measurements of direct and diffuse solar radiations are used to measure spectral AODs and retrieve SSAs and other aerosol optical-microphysical properties (volume size distribution, refractive index, phase function, and asymmetry parameter) at the same standard wavelengths of AOD following an inversion algorithm packaged in the *SKYRAD.pack* software (*Nakajima et al.*, 1996; *Hashimoto et al.*, 2012). Cloudy observations are screened using the Cloud Screening Sky Radiometer code (*Khatri and Takamura*, 2009).

The SKYNET radiometers come in two flavors, model POM-01 and model POM-02. The POM-01 instrument carries a total of five wavelength filters covering visible to near-IR (400-1020 nm), whereas POM-02 instrument has two additional filters in the UV region (340 and 380 nm) along with the other filters in the visible to shortwave-IR (including 1627 nm and 2200 nm) part of the spectrum. The calibration of each SKYNET radiometer is performed on-site on a monthly basis using the improved Langley method (*Nakajima et al.*, 1996, *Campanelli et al.*, 2004, 2007). Occasionally, the inter-calibration of radiometers is carried out against the master instrument well-calibrated using the Langley method at a high mountain site, e.g., Mauna Loa. The SKYNET radiometers are also inter-compared with AERONET Cimel Sunphotometers and precision filter radiometers at three observation sites, i.e., *Chiba University* and *Valencia* (*Estelles et al.*, 2016), and *Rome* (*Campanelli et al.*, 2018) .

Studies in the past have compared AODs (*Estellés et al.*, 2012a) and SSAs (*Estellés et al.*, 2012b) measured/retrieved from SKYNET and AERONET and shown that AODs are well-correlated and in good agreement, but the SKYNET SSAs are found to be higher than those of AERONET (*Che et al.*, 2008; *Hashimoto et al.*, 2012). *Khatri et al.* (2016) further pinpoints the factors, such as quality of input data attributed to different calibration and observation protocol, different



quality assurance criteria, the calibration constant for sky radiances, differences in measured
AOD, and surface albedo, responsible for the inconsistent aerosol SSA between AERONET and
SKYNET using observations from the four representative sites, i.e., Chiba (Japan), Pune (India),
Valencia (Spain), and Seoul (South Korea). More discussion on the sources of uncertainties is
presented in section 4.

In this study, we include the SKYNET data acquired over a total of 25 sites distributed mostly
across Asia and a few in Europe. The dataset is freely accesible from the data holding portal of
the Center for Environmental Remote Sensing (CERes), Chiba University, Japan
(http://atmos3.cr.chiba-u.jp/skynet/data.html). Figure 1 shows the geographic distribution of
selected sites, whereas Table 1 lists the geo-coordinates of these sites with the associated
sensor type (POM-01 or POM-02) and data periods. The SKYNET aerosol product is derived
using two different Skyrad packs: version 4.2 and version 5, the differences of which are
explained in *Hashimoto et al*. (2012). In this study, we use the SKYNET Level 2 product retrieved
using version 5 of Skyrad pack. SKYNET retrievals assigned with cloud flag '0' are included in the
analysis, since these measurements are  believed to be free of cloud contamination considered
as higher quality retrievals. A careful examination of the SKYNET inversion dataset revealed
some irregularities in the measurements for many sites, such as irregular patterns in the shape
of spectral SSAs, identical values of SSA at near-UV and visible wavelengths, and much larger
standard deviation (>0.1) in SSA within a few hours. These spurious measurements were
excluded from the present analysis.
**2.3   THE COLLOCATION OF OMI AND SKYNET MEASUREMENTS**
OMI retrievals correspond to a spatial scale of 13 × 24 km$^2$ at nadir representing the
atmospheric conditions over an area. Unlike the direct measurements of the spectral AOD,
which correspond to columnar point measurements, the retrievals made by SKYNET use the sky
radiances measured at several discrete angles azimuthally, therefore representing the sky
condition observed over a station which is associated with approximately 5 km radius
surrounding the Sun photometer site. SKYNET retrieves aerosol optical-microphysical



properties, including spectral SSA, under all cloud-free conditions and at all aerosol loadings. It
is expected that the inversion of retrieved parameters from sky radiances offers better accuracy
at larger solar zenith angles owing to the longer optical path and better aerosol absorption
signal (*Dubovik et al.*, 2000). These conditions are best satisfied with the measurements made
during the early morning and late afternoon hours. On the other hand, Aura/OMI overpasses a
station during the afternoon hours with the local equator-crossing time 1:30 P.M. In order to
collocate both types of measurements, therefore we select a time window of ±3 h around OMI
overpass time in order to get sufficient high-quality SKYNET retrievals particularly from early
morning/late afternoon measurements. The OMI retrievals of SSA were spatially averaged in a
grid area of 0.5° by 0.5° centered at the SKYNET site. Though the spatial averaging area for the
OMI retrieval is about 50 km$^2$, due to its larger footprint, the actual area intercepted by OMI
pixels around SKYNET site is likely to be larger.

OMI performs retrieval at 354 nm and 388 nm wavelengths, whereas the SKYNET POM-02
instrument reports SSA at nearby wavelengths of 340, 380, and 400 nm. To compare both SSA
products at the same wavelength, SKYNET SSA was linearly interpolated at 388 nm, to match
with the wavelength of OMI retrieval, using the measurements at the two nearest wavelengths,
i.e., 380 nm and 400 nm. The SKYNET POM-01 instruments don't carry UV wavelength filters,
but report the retrievals at the shortest wavelength 400 nm and other visible/near-IR
wavelengths. In this case, the OMI retrievals are extrapolated from 388 nm to 400 nm, to match
with the wavelength of SKYNET inversion, following the spectral dependence of SSA associated
with the chosen aerosol model in the OMI algorithm.
**3  RESULTS**
**3.1  OMI-SKYNET COMPARISON OVER INDIVIDUAL STATIONS**
Figure 2 displays the OMAERUV versus SKYNET SSA scatterplots for selected sites in Japan. The
comparison was made at 388 nm or 400 nm depending upon the availability of the SKYNET
inversion at those wavelengths, i.e., POM-01 or POM-02 sensors. Legends with different colors



represent the aerosol type selected by the OMAERUV algorithm for the co-located matchups
(N). RMSD is the root-mean-square difference between the two retrievals; Q_0.03 and Q_0.05
are the percent of total matchups (N) that fall within the absolute difference of 0.03 and 0.05,
respectively; horizontal and vertical lines for each matchup are the standard deviation of
temporally and spatially averaged SKYNET and OMI SSAs. The comparison includes OMI-SKYNET
matchups with AOD>0.3 (388 or 400 nm) in both measurements simultaneously. The
scatterplots reveal a good level of agreement for matchups identified with carbonaceous
aerosols over *Chiba University*, *Cape Hedo*, *Fukue*, *Saga*, and *Etchujima* with the majority of
points confined within the absolute difference of 0.03. The OMI-SKYNET dataset are dominated
with matchup points identified as the urban/industrial aerosols by the OMAERUV algorithm for
which the measured UVAI falls below 0.5 representing lower aerosol loading in the boundary
layer with weakly absorbing properties. Under such observed conditions, the uncertainties in
both kinds of measurements are prone to be larger due to lower absorption signal relative to
the instrumental noise and errors in algorithmic assumptions, such as surface albedo, that
could further amplify the overall uncertainty in the retrievals. Despite these inherent
uncertainties, an agreement within the difference of ±0.03 for about or more than half of the
collocated retrievals is encouraging. A more detailed description of the different sources of
uncertainty is presented in the next section.

Figure 3 shows the scatterplots of OMI-SKYNET SSA for remaining sites located in South Korea,
China, Thailand, India, and Italy. For the site *Seoul* in South Korea, OMI tends to overestimate
SSA for a number of matchups assigned with the urban/industrial aerosol type and for a few
with the carbonaceous/smoke aerosol type such that about 42% of total matchups are falling
within the difference of 0.03. For the *Dunhuang* site located in the desert area of China, a
majority of collocated data points were identified as dust aerosol type providing an overall
better agreement with 50% and 68% matchups bounded within ±0.03 and ±0.05 differences,
respectively. The *Phimai* site in Thailand is known to be influenced by the springtime biomass
burning activities, where OMI and SKYNET SSAs are found to agree relatively best among all 25
sites providing 71% and 91% of the matchups restricted within ±0.03 and ±0.05 limits,



respectively. The agreement between the two sensors was robust for the carbonaceous/smoke
aerosol type followed by the urban/industrial aerosols. Over the megacity of New Delhi in the
Indo-Gangetic Plain in India, which is seasonally influenced by the smoke and desert dust
aerosols in addition to the local source of urban pollution, the OMI-SKYNET matchups are found
to agree within ±0.03 and ±0.05 for 52% and 83% of the evaluated data points respectively.
Over the *Pune* station located near the western boundary of India and the *Bologna* site in Italy,
OMI retrieves higher SSA compared to that of SKYNET yielding 39% and 64%, and 25% and 50%
matchups, respectively, within the two uncertainty limits. Table 1 lists the statistical measures
of the OMI-SKYNET SSA comparison for all 25 sites.
**3.2  COMPOSITES FOR EACH AEROSOL TYPE**
Figure 4 displays the composite scatterplots of OMI versus SKYNET SSA derived by segregating
the matchup points for each aerosol type from all 25 sites. The intention here is to evaluate the
consistency between the two retrieval methods for each aerosol type separately and
understand their relative differences. When identified as the carbonaceous/smoke aerosol type,
the OMI-SKYNET matchups reveal relatively best comparison among the three major aerosol
types with 61% and 84% data points falling within the absolute difference of 0.03 and 0.05,
respectively, and providing the lowest (0.035) root-mean-square-difference between the two
retrievals. The collocation procedure yields the lowest number of matchups (N=32) for desert
dust aerosol type obtained mostly over the site of *Dunhuang* in China, resulting 50% and 72% of
data points within the stated uncertainty limits. Among the three aerosol types, the collocated
points assigned with the urban/industrial aerosol type (Figure 4 bottom-left) yield the
maximum number of matchups (N=739) with the relatively weakest agreement (RMSD=0.052),
where OMI tends to overestimate SSA for a significant number of instances resulting about 45%
and 67% data points falling within the two limits of expected uncertainties. When more than
one prescribed aerosol types are selected for OMI pixels around the SKYNET stations, the
matchups between the two sensors resulted in 59% and 77% retrievals within the uncertainty
limits with an RMSD of 0.041—a comparison slightly poorer than 'smoke-only' case, but better
than 'dust-only' and 'urban/industrial-only' retrieval cases. Combined, all three distinct aerosol



types simultaneously yield the total number of matchups (N=1223) with an RMSD of 0.047
between OMI and SKYNET resulting 51% and 72% collocated data points falling within the
absolute difference of 0.03 and 0.05 difference, respectively. When the restriction of AOD>0.3
is removed from the collocation procedure, allowing all matchups regardless of their respective
AOD values, the total number of collocated data points was increased to more than twice
(N=2691) albeit with a relatively weaker agreement yielding an RMSD of 0.06 and percent data
points within the uncertainty limits reducing to 38% and 59%, respectively.
**3.3  DIAGNOSIS OF OMAERUV VERSUS SKYNET**
The SKYNET algorithm inverts the spectral sky radiances in conjunction with the direct AOD
measurements to retrieve the real and imaginary parts of the refractive index and particle size
distribution of cloud-free observations under all aerosol loading conditions. These inversion
products are believed to be more stable and accurate at larger aerosol loadings and solar zenith
angles due to stronger aerosol absorption signal and longer optical path  (*Dubovik et al.*, 2000).
Similarly, a sensitivity analysis of the two-channel OMAERUV retrievals suggests that the
retrieved AOD and SSA are susceptible to the small change in surface albedo at lower aerosol
loading (*Jethva et al.*, 2014). For instance, an absolute difference of 0.01 in the surface albedo
leads to a change in AOD approximately by 0.1 and SSA by ~0.02.

Figure 5 (top) shows the absolute difference in collocated SSA between OMI and SKYNET as a
function of concurrent SKYNET direct AOD (388 or 400 nm) measurements for all aerosol types.
All OMI-SKYNET matchup data obtained from a total of 25 sites under all AOD conditions are
included here. The data are shown in the box and whisker format, where the horizontal lines
represent the median value of each bin of sample size 150, filled circle the mean value, and
shaded vertical bars cover the 25 and 75 percentiles of the population in each bin. While for
most bins the mean and median values of SSA difference were restricted to within ±0.03, OMI
tends to overestimate SSA relative to that of SKYNET at lower AODs giving larger differences
and spread in the data population. Similar patterns were observed when the difference in SSA
was related to the OMI-retrieved AOD (Figure 5 middle). In both cases, the differences in SSA



minimize at larger AOD values (>0.5) suggesting a convergence in both retrievals. Figure 5
(bottom) shows a similar plot of SSA difference against the concurrent OMI UVAI. Notably, the
differences in SSA exhibit even a stronger relationship to UVAI than that in the AOD case (top
and middle). For UVAI lesser than zero, the differences in the retrieval are found to be beyond
the expected uncertainty in both inversions, at least in the mean sense. For the lower range of
UVAI, OMI algorithm mostly employs the urban/industrial model for the retrieval where all
aerosols are assumed to be confined within the boundary layer (<2 km) with a vertical profile
that follows an exponential distribution. On the other hand, the mean and median values of the
SSA difference for UVAI larger than 0.2 for all bins fall within the 0.03 uncertainty range. The
SSA differences approach to near-zero with a reduced spread at larger magnitudes. Notably,
both inversions are found to be in closer agreement for UVAI measurements>0.3.

**4   SOURCES OF UNCERTAINTY**
**4.1   UNCERTAINTIES IN THE GROUND-BASED SKYNET INVERSION PRODUCT**
The SKYNET inversion algorithm assumes a wavelength-independent surface albedo of 0.1 at all
wavelengths across the UV to visible part of the spectrum. The diffuse light reflected from the
ground plays a second-order role in the measured sky radiances in most situations, however,
has a potential to affect the SSA inversion, e.g., overestimated (underestimated) surface albedo
can underestimate (overestimate) SSA (*Dubovik et al.*, 2000; *Khatri et al.*, 2012). Using
simultaneous inversion data from SKYNET and AERONET for four representative sites, *Khatri et*
*al.* (2016) have shown that the difference in the prescribed surface albedo between SKYNET
and AERONET results in a difference of ~0.04 in SSA at red (675 nm) and near-IR wavelengths
retrieved from the two collocated ground sensors. The difference in SSA can also reach as large
as ~0.08 when surface albedo differed by 0.3. The assumed surface albedo value of 0.1 at near-
UV (340 and 380 nm) and shorter visible wavelength (400 nm) seems to be unrealistic for the
vegetated and urban surfaces. The surface albedo database at 354 nm and 388 nm derived
from multiyear observations from OMI suggests that the vegetated surfaces and urban centers





are characterized with the lower values of surface albedo, i.e., ~0.02-0.03 and ~0.05,
respectively; for desert surfaces, the albedo could be as high as 0.08-0.10. Significant
differences in the assumed surface albedo values between OMI and SKYNET at shorter
wavelengths could be one of the responsible factors for discrepancies in SSA noted over several
sites, particularly at lower aerosol loading when the uncertainty in surface characterization can
amplify error in the SSA inversion.

To further investigate this effect, the difference in SSA between OMI and SKYNET as a function
of the simultaneous difference in surface albedo is analyzed and shown in Figure 6. The data
are presented in a standard box and whisker plot format. Figure 6 reveals a link between
differences in SSA and surface albedo, where increasing differences in SSA (OMI>SKYNET) are
associated with significant negative biases in surface albedo between OMI and SKYNET. In other
words, large overestimation in SKYNET surface albedo causes underestimation of retrieved SSA,
which is consistent with the findings of *Dubovik et al*. (2000) and *Khatri et al*. (2012, 2016),
thereby resulting in a substantial positive difference in SSA between OMI and SKYNET. Recently,
*Mok et al.* (2018) have shown that the use of AERONET surface albedo dataset at 440 nm in the
SKYNET algorithm for the S. Korea region produces SSA values larger by ~0.01 at near-UV
wavelengths.  Notably, differences in SSA tend to be lower when the differences in surface
albedo are also minimal, such that the mean and median values of those bins remain within the
expected uncertainties of ±0. 03 in both retrievals. This result, along with the previous findings
cited above, convincingly points out that the SSA inversion from ground-based sensors,
especially at lower aerosol loadings, is likely susceptible to the prescribed surface albedo. The
assumption of a fixed value of spectral surface albedo of 0.1 in the SKYNET algorithm appears to
be inappropriate calling for a revision using more accurate datasets of spectral reflectance or
albedo such as from MODIS and OMI.

SKYNET inversion algorithm (Skyrad.pack Version 4.2 and version 5) assumes aerosols of
spherical shape regardless of the actual aerosol type observed in the scene. Following a



detailed analysis of the effect of non-sphericity of the particles on the difference between the
retrievals carried out assuming spherical and spheroidal size distribution, *Khatri et al.*, (2016)
concluded that the assumed shape of particles has a non-significant impact on the retrieved
SSA. Their study revealed SSA difference of ±0.01 for measurements having a maximum
scattering angle <120° and difference of up to ±0.02 at scattering angle >120°, where the
difference in the phase function is significant between spherical and spheroidal size
distributions (*Torres et al.*, 2018).  The OMI-SKYNET collocation procedure, as shown in Figure 4,
yields relatively fewer matchups that are identified as dust aerosol type according to the
OMAERUV aerosol type identification scheme. A majority of the collocated data points were
derived over the desert site of *Dunhuang* in China showing a reasonable agreement in SSA
between OMI and SKYNET for dust aerosols further supporting the findings of *Khatri et al*.
(2016) that the SSA retrievals are not significantly impacted by the assumption of the shape of
particles, i.e., spherical or spheroidal.

Apart from the algorithmic assumptions, the calibration constant used for sky radiances
measured by SKYNET instruments can be a potential source of errors in the inversion. *Khatri et*
*al*. (2016) suggests that the calibration constant for sky radiances determined from the disk
scan method using solar disk scan area of 1° × 1° (*Boi et al.*, (1999) may be underestimated
resulting in overestimated sky radiance and thus relatively higher SSA. Some of the larger
differences between in SSA between OMI and SKYNET, where OMI underestimates SSA relative
to the SKYNET, can be attributed to the imperfect calibration applied to the SKYNET sensors.
**4.2  POSSIBLE SOURCES OF UNCERTAINTIES IN OMAERUV RETRIEVALS**
Like other satellite-based remote sensing algorithms, OMAERUV also relies on assumptions
about the atmospheric and surface properties for the retrieval of aerosol properties. The single
largest known source of error in the OMI retrievals is the subpixel cloud contamination within
the OMI footprint. Given the footprint of size 13 × 24 $km^2$ for near-nadir pixels which intercept
an area of about 338 $km^2$ on the ground, the presence of subpixel clouds may not be avoided
entirely. Currently, the algorithm assigns quality flags to each pixel which carries information on





the quality of the retrieval depending upon the observed conditions (*Torres et al.*, 2013).
Aerosol retrieval with the quality flag '0' are considered to be the best in accuracy as this
category of flag scheme largely avoids cloud-contaminated pixels by choosing the appropriate
thresholds in reflectivity and UVAI measurements.

Over the desert regions, e.g., the *Dunhuang* SKYNET site in China, the frequency of occurrence
of clouds is expected to be minimal. Therefore, it is less likely that the SSA retrievals over these
sites are affected by cloud contamination. A reasonable agreement between the two retrievals
(Figure 3) supports this assumption. The quality flag scheme, however, cannot entirely rule out
the presence of small levels of subpixel cloud contamination or the presence of thin cirrus in
the OMI footprint, which can cause overestimation in the retrieval of SSA, such as noted over
the SKYNET sites in *Kasuga*, *Etchujima*, *Seoul*, *Bologna*, and *Pune*.

Another possible source of uncertainty can be the assumption of the aerosol layer height. The
climatology of aerosol layer height derived from CALIOP measurements adequately describes
the observed mean layer of carbonaceous and desert dust aerosols (*Torres et al.*, 2013). It is
particularly robust over the arid and semiarid areas where large numbers of cloud-free
observations were used in the calculation. However, note that the temporal and spatial
coverage of CALIOP is limited to 16-day repeat cycle over the same location. Variations in the
aerosol layer height not observed by CALIOP, therefore, will be missed out in the derived
climatology and thus can be a source of uncertainty. Sensitivity analysis of the OMAERUV
retrievals suggests that an overestimation (underestimation) in the aerosol layer height results
in an overestimated (underestimated) SSA. This is because an increase (decrease) in the aerosol
layer height from the actual one enhances (reduces) absorption signal in the radiance
measurements in near-UV, which the OMAERUV algorithm compensates by retrieving higher
AOD and SSA to match with the observations.



The third source of uncertainty that can affect SSA retrieval is the accuracy of the prescribed
surface albedo. For the surface characterization, the OMAERUV algorithm use a near-UV
surface albedo database derived using the multiyear OMI reflectivity observations. The method
adopts a minimum reflectivity approach, ensuring minimal or no contamination from the
atmosphere, i.e., aerosols and clouds, in the measurements. Afterward, the minimum
reflectivity dataset derived from the OMI observations was adjusted in the temporal domain to
the seasonality of surface albedo retrieved in the visible wavelengths from MODIS. The dataset
contains surface albedo values at 354 and 388 nm at a grid resolution of 0.25° × 0.25°.
Compared to the previous OMAERUV dataset using TOMS-based surface albedo product at 1°
grid resolution, the new OMI-based dataset is expected to be more accurate to within 0.005 to
0.01 owing to its higher spatial resolution and the fact that it is contemporary to the OMI
operation. A sensitivity study of the OMAERUV retrievals to the change in surface albedo
described in *Jethva et al*. (2014) suggests that an increase in surface albedo by 0.01 in the near-
UV region over desert areas results in a decrease in the magnitude of retrieved SSA by ~ -0.02.
The effect of uncertain surface albedo can be more pronounced at lower aerosol loading,
where the reduced signal from the atmosphere makes OMAERUV retrieval more susceptible to
the uncertainty in surface albedo.

The assumed aerosol microphysical and optical properties could be additional sources of
uncertainty. The particle size distributions assumed in the OMAERUV models are adopted from
long-term AERONET inversion statistics (*Dubovik et al*., 2002), representing areas influenced by
smoke, dust, and urban/industrial aerosols, and therefore are considered realistic
representations of the total atmospheric column. The carbonaceous smoke aerosols are
assumed to be spherical in shape with a bimodal log-normal size distribution and characterized
with a steep absorption gradient, such that the Absorption Angstrom Exponent (AAE) in the
near-UV lies in the range 2.5-3.0, to adequately represent the organics in the biomass burning
smoke particles (*Kirchstetter et al*., 2004; *Jethva and Torres*, 2011). The desert dust aerosol
model follows bimodal log-normal size distribution with particles comprised of randomly
oriented spheroids with an axis ratio (shape factor) distribution adopted from *Dubovik et al*.



(2006). The spectral dependence of the refractive index in the near-UV assumed in the dust
aerosol model is generally consistent with the in-situ laboratory measurements (*Wagner et al.*,
2012). For instance, retrieval of AOD and SSA for carbonaceous aerosols using the smoke model
with AAE of 1.90 (10% relative spectral dependence in the imaginary index between 354 and
388 nm) and 1.0 (no spectral dependence in the imaginary index), instead of the standard AAE
assumption of 2.7, results in a decrease in SSA up to -0.07, respectively, suggesting a marked
sensitivity of the SSA retrieval to the significant changes in the spectral aerosol absorption. Due
to the shortage of ground-based characterization of absorption in the near-UV part of the
spectrum, the regional representation of the spectral absorption properties in the OMAERUV
models is limited. Therefore, spatial and temporal variations in the actual aerosol spectral
properties can be a potential source of error in the SSA retrieval.

## 5    SUMMARY AND CONCLUSION

We presented a comparative analysis of the aerosol SSA retrieved from the OMI's two-channel
aerosol algorithm (OMAERUV) against an independent ground-based inversion made by the
SKYNET Sun photometers over selected 25 sites located mainly in Asia and Europe. This study
follows our previous efforts of evaluating the OMI near-UV SSA product carried out using
ground-based AERONET dataset (*Jethva et al.*, 2014). The capability of SKYNET sensors to
measure the Sun and sky radiance at near-UV wavelengths (340-380-400 nm), and
subsequently retrieve the aerosol optical properties, including SSA, at these wavelengths
provide a unique opportunity to directly compare the two near-UV SSA products from ground
and satellite. Ground-based inversion of SSA at the near-UV wavelengths eliminate the need to
adjust and extrapolate satellite retrieval to the visible wavelengths such as the case with
comparison against AERONET.   Since the SSA inferred from two different platforms are
essentially retrieved from two fundamentally different inversion algorithms, the present study
does not stand as a "validation" exercise for either retrieval data sets. Instead, the purpose of
this analysis was to check the consistency (or lack thereof) between the two retrieved



quantities of the same physical parameter regarding standard statistical comparison, i.e., RMSD
and % of matchups within the expected uncertainties.

Unlike AERONET Level 2 inversion product that reports spectral SSA when AOD (440 nm)
exceeds a value of 0.4, SKYNET Level 2 dataset delivers spectral SSA in the near-UV and visible
parts of the spectrum under all cloud-free observations for all AOD conditions. The collocation
procedure that matched temporal inversion data from SKYNET with spatial retrievals from OMI
gave resulted in a total of 2691 collocated data points for AOD>0.0 and 1223 when AOD>0.3
collected from 25 sites representing biomass burning region of Southeast Asia, desert in China,
and urban/industrial areas in Japan, India, and Europe. Combinedly for all 25 sites and under all
AOD conditions, we find 38% and 59% of the total SKYNET-OMI SSA agree within their
estimated uncertainty range of ±0.03 and ±0.05, respectively, with an overall root-mean-
square-difference of 0.06. When restricted with condition AOD>0.3 in both measurements, the
agreement of comparison improved to 51% and 72% with root-mean-square-difference of
0.047. When segregated by aerosol type, the agreement between the two sensors is found to
be robust for matchups identified as the carbonaceous aerosols over several sites in Japan,
*Seoul* in South Korea, *Phimai* in Thailand, and *New Delhi* in India, yielding 61% and 84% of data
points falling within the limits of ±0.03 and ±0.05 with an overall RMSD of 0.035. The
collocation procedure found few matchups for desert dust aerosol, mostly over *Dunhuang* site
in China, showing a reasonable comparison with 50% and 68% data points within expected
uncertainty limits. Among the three major aerosol types, the urban/industrial type aerosols
provide the maximum number of matchup data points with a relatively poorer comparison,
where 45% and 67% data are found to be within the uncertainty limits.

The differences in SSA between OMI and SKYNET are found to be largest at lower aerosol
loading, where OMI retrieves significantly higher SSA compared to that of SKYNET. However,
the differences are minimized at larger AOD values (>0.5) suggesting a convergence in both
retrievals at moderate to large aerosol loading. Similarly, the differences in SSA exhibit a



stronger relationship to UVAI showing larger discrepancies beyond expected uncertainty limits
at lower UVAIs (<0), but nearing to zero with a reduced spread in matchups at larger
magnitudes of UVAI (>0.2-0.3).

Much of the inconsistency observed between OMI and SKYNET at lower aerosol loadings
indicate retrieval issues due to reduced signal-to-noise ratio and uncertain algorithmic
assumptions. For instance, the OMAERUV retrievals are more susceptible to the changes in
surface albedo at lower AODs, and to the spectral absorption at higher AODs (*Torres and Jethva*,
2011). On the other hand, the SKYNET inversion algorithm assumes a wavelength-independent
surface albedo of 0.1 across the UV to visible-near-IR wavelengths, which appears to be
unrealistic especially in the UV region where OMI surface albedo dataset shows much lower
values (<0.05) over land. Though the reflected light from surface plays a second-order role in
the ground-based retrievals, previous studies as well as shown in the present work (Figure 6),
uncertainty in surface albedo can cause non-negligible errors in SSA retrievals that likely exceed
the expected accuracy level of ±0.03.

Despite the inherent uncertainties associated with both satellite and ground inversion products,
a good level of agreement between the two independent techniques over SKYNET sites at
increasing aerosol loading is encouraging. We intend to extend the present analysis to other
SKYNET sites whose data are still not directly accessible in the public domain. Continuing the
evaluation of inversion products, both from satellite and ground, is an important exercise to
track the changes and improvements in the algorithms and resulting data products, and to
establish the consistency (or lack thereof) that can help to diagnose further and improve the
accuracy of retrievals.



## ACKNOWLEDGMENTS

We thank the Center for Environmental Remote Sensing (CERes), Chiba University, Japan (http://atmos3.cr.chiba-u.jp/skynet/data.html), for the online availability of the SKYNET dataset for several sites in Japan, South Korea, China, India, Italy, and Germany. Acknowledgments are also due to the principal investigators and their staff for establishing and maintaining respective SKYNET sites, whose data are used in the present work. We acknowledge the support of NASA GES-DISC, the NASA Earth Science data center, for the online availability of the OMI aerosol product assessed in this analysis.



**AUTHORS' CONTRIBUTIONS**
Dr. Jethva, the leading author, conceptualized the study and wrote the paper. He conducted
comparative data analysis of OMI- and SKYNET-retrieved single-scattering albedo products
presented in the paper. Dr. Torres (2nd author) brought his expertise in interpreting the results
and helped improving the manuscript writeup.

**Additional Information**
The author(s) declare no competing interests, financial or non-financial.



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

Preliminary aerosol optical depth comparison between ESR/SKYNET, AERONET and GAW
international networks. International SKYNET workshop, Rome (Italy), 2016.
Khatri, P., and T. Takamura: An algorithm to screen cloud affected data for sky radiometer data
analysis, J. Meteor. Soc. Japan, 87, 189-204, 2009.
Khatri, P., T. Takamura, A. Yamazaki, and Y. Kondo: Reterival of key aerosol optical parameters
for spectral direct and diffuse irradiances measured by a horizontal surface detector, J. Atmos.
Oceanic Technol., 29, 683–696, 2012.
Khatri, P., T. Takamura, T. Nakajima, V. Estellés, H. Irie, H. Kuze, M. Campanelli, A. Sinyuk, S.-M.
Lee, B. J. Sohn, G. Pandithurai, S.-W. Kim, S. C. Yoon, J. A. Martinez-Lozano, M. Hashimoto, P. C.
S. Devara, and N. Manago: Factors for inconsistent aerosol single scattering albedo between
SKYNET and AERONET, J. Geophys. Res. Atmos., 121, 1859-1877, doi:10.1002/2015JD023976,
642    2016.





Kirchstetter, T. W., T. Novakov, and P. V. Hobbs: Evidence that the spectral dependence of light
absorption by aerosols is affected by organic carbon, J. Geophys. Res., 109, D21208,
doi:10.1029/2004JD004999, 2004.

Hansen, J., M. Sato, and R. Ruedy: Radiative forcing and climate response, J. Geophys. 795 Res.,
102(D6), 6831-6864, doi:10.1029/96JD03436, 1997.


Hashimoto, M., Nakajima, T., Dubovik, O., Campanelli, M., Che, H., Khatri, P., Takamura, T., and
Pandithurai, G.: Development of a new data-processing method for SKYNET sky radiometer
observations, Atmos. Meas. Tech., 5, 2723-2737, https://doi.org/10.5194/amt-5-2723-2012,
655   2012.


IPCC, 2013: Climate Change 2013: The Physical Science Basis. Contribution of Working Group I
to the Fifth Assessment Report of the Intergovernmental Panel on Climate Change (Stocker, T.F.,
D. Qin, G.-K. Plattner, M. Tignor, S.K. Allen, J. Boschung, A. Nauels, Y. Xia, V. Bex and P.M.
Midgley (eds.)). Cambridge University Press, Cambridge, United Kingdom and New York, NY,
USA, 1535 pp, doi:10.1017/CBO9781107415324.

Jethva, H., and O. Torres: Satellite-based evidence of wavelength-dependent aerosol absorption
in biomass burning smoke inferred from Ozone Monitoring Instrument, Atmos. Chem. Phys., 11,
10,541–10,551, doi:10.5194/acp-11-10541-2011, 2011.

Jethva, H., O. Torres, and C. Ahn: Global assessment of OMI aerosol single-scattering albedo
using ground-based AERONET inversion, J. Geophys. Res. Atmos., 119,
doi:10.1002/2014JD021672, 2014.

Mok, J., Krotkov, N. A., Torres, O., Jethva, H., Li, Z., Kim, J., Koo, J.-H., Go, S., Irie, H., Labow, G.,
Eck, T. F., Holben, B. N., Herman, J., Loughman, R. P., Spinei, E., Lee, S. S., Khatri, P., and
Campanelli, M.: Comparisons of spectral aerosol single scattering albedo in Seoul, South Korea,
Atmos. Meas. Tech., 11, 2295-2311, https://doi.org/10.5194/amt-11-2295-2018, 2018.

Nakajima, T., G. Tonna, R. Rao, P. Boi, Y. Kaufman, and B. Holben: Use of sky brightness
measurements from ground for remote sensing of particulate polydispersions, Appl. Opt., 35,
678   15, 2672-2686, 1996.



Schenkeveld, V. M. E., Jaross, G., Marchenko, S., Haffner, D., Kleipool, Q. L., Rozemeijer, N. C.,
Veefkind, J. P., and Levelt, P. F.: In-flight performance of the Ozone Monitoring Instrument,
Atmos. Meas. Tech., 10, 1957–1986, https://doi.org/10.5194/amt-10-1957-2017, 2017.
Torres, O., P. K. Bhartia, J. R. Herman, Z. Ahmad, and J. Gleason: Derivation of aerosol
properties from satellite measurements of backscattered ultraviolet radiation: Theoretical basis,
J. Geophys. Res., 103(D14), 17,099–17,110, doi:10.1029/98JD00900, 1998.
Torres, O., P. K. Bhartia, A. Sinyuk, E. J. Welton, and B. Holben:Total Ozone Mapping
Spectrometer measurements of aerosol absorption from space: Comparison to SAFARI 2000
ground-based observations, J. Geophys. Res., 110, D10S18, doi:10.1029/2004JD004611, 2005
Torres, O., A. Tanskanen, B. Veihelmann, C. Ahn, R. Braak, P. K. Bhartia, P. Veefkind, and P.
Levelt: Aerosols and surface UV products from Ozone Monitoring Instrument observations: An
overview, J. Geophys. Res., 112, D24S47, doi:10.1029/2007JD008809, 2007.
Torres, O., C. Ahn, and Z. Chen: Improvements to the OMI near-UV aerosol algorithm using A-
train CALIOP and AIRS observations, Atmos. Meas. Tech., 6, 3257–3270, doi:10.5194/amt-6-
696 3257-2013, 2013.

Torres, O., Bhartia, P. K., Jethva, H., and Ahn, C.: Impact of the ozone monitoring instrument
row anomaly on the long-term record of aerosol products, Atmos. Meas. Tech., 11, 2701-2715,
https://doi.org/10.5194/amt-11-2701-2018, 2018.
Wagner, R., T. Ajtai, K. Kandler, K. Lieke, C. Linke, T. Müller, M. Schnaiter, and M. Vragel:
Complex refractive indices of Saharan dust samples at visible and near UV wavelengths: A
laboratory study, Atmos. Chem. Phys., 12, 2491–2512, doi:10.5194/acp-12-2491-2012, 2012.





**TABLES**
*Table 1 A list of SKYNET sites and corresponding dataset used in the present analysis. Sensor*
*type "POM02" consists of a total of seven wavelength filters, including near-UV bands, i.e., 340,*
*380, 400, 500, 675, 870, and 1020 nm, whereas "POM01" sensors have a total of five*
*wavelength filters, i.e., 400, 500, 675, 870, and 1020 nm. The rightmost four columns enlist the*
*statistical measures of OMI-SKYNET single-scattering albedo matchups.*
*Abbreviations: N: number of satellite-ground matchups, RMSD: root-mean-square-difference between OMI and*
*SKYNET, Q_0.03 and Q_0.05: percent matchups within an absolute difference of 0.03 and 0.05.*

| SKYNET Station Name | Longitude | Latitude | Country | Sensor Type | Data Period | N | RMSD | Q_0.03 (%) | Q_0.05 (%) |
|---|---|---|---|---|---|---|---|---|---|
| *Chiba University* | 140.104°E | 35.625°N | Japan | POM02 | 2005-2017 | 132 | 0.039 | 58 | 81 |
| *Cape Hedo* | 128.248E | 26.867N | Japan | POM02 | 2005-2017 | 47 | 0.044 | 47 | 72 |
| *Fukue* | 128.682E | 32.752N | Japan | POM02 | 2008-2017 | 71 | 0.041 | 59 | 76 |
| *Miyako* | 125.327E | 24.737N | Japan | POM02 | 2004-2017 | 31 | 0.059 | 23 | 58 |
| *Sendai* | 140.84E | 38.26N | Japan | POM01 | 2009-2017 | 34 | 0.052 | 50 | 74 |
| *Kasuga* | 130.475E | 33.524N | Japan | POM02 | 2004-2017 | 159 | 0.057 | 40 | 61 |
| *Saga* | 130.283E | 33.233N | Japan | POM02 | 2011-2017 | 66 | 0.044 | 52 | 71 |
| *Minamitorishima* | 153.97E | 24.3N | Japan | POM02 | 2006-2009 | - | - | - | - |
| *Moshiri* | 142.260E | 44.366N | Japan | POM02 | 2009-2011 | 2 | 0.018 | 100 | 100 |
| *Fuji Hokuroku* | 138.750E | 35.433N | Japan | POM02 | 2009-2017 | 9 | 0.051 | 56 | 67 |
| *Tsukuba* | 140.096E | 36.114N | Japan | POM02 | 2014-2017 | 5 | 0.027 | 80 | 100 |
| *Takayama* | 137.423E | 36.145N | Japan | POM02 | 2014-2017 | 3 | 0.022 | 67 | 100 |
| *Etchujima* | 139.796E | 35.664N | Japan | POM01 | 2004-2010 | 100 | 0.052 | 45 | 66 |
| *Seoul* | 126.95E | 37.46N | Republic of South Korea | POM01 | 2005-2015 | 182 | 0.050 | 42 | 66 |
| *Yonsei* | 126.980E | 37.570N | Republic of South Korea | POM02 | 2016 | 5 | 0.035 | 40 | 80 |
| *Dunhuang* | 90.799E | 40.146N | China | POM01 | 1999-2007 | 40 | 0.048 | 50 | 68 |
| *Phimai* | 102.564E | 15.184N | Thailand | POM02 | 2005-2017 | 139 | 0.031 | 71 | 91 |
| *Bangkok* | 100.605E | 13.667N | Thailand | POM02 | 2009-2017 | 15 | 0.064 | 47 | 60 |
| *Mandalgovi* | 106.264E | 45.743N | Mongolia | POM01 | 1998-2009 | 4 | 0.087 | 0 | 0 |
| *Ulan Bator* | 106.921E | 47.923N | Mongolia | POM01 | 2013-2017 | 2 | 0.026 | 100 | 100 |
| *New Delhi* | 77.174E | 28.629N | India | POM01 | 2006-2007 | 63 | 0.038 | 52 | 83 |
| *Pune* | 73.805E | 18.537N | India | POM01 | 2004-2009 | 94 | 0.050 | 39 | 64 |
| *Bologna* | 11.34E | 44.52N | Italy | POM02 | 2014-2017 | 114 | 0.065 | 25 | 50 |
| *Valencia* | 0.420E | 39.507N | Spain | POM01 | 2014-2017 | 4 | 0.052 | 25 | 25 |
| *Bremen* | 8.854E | 3.108N | Germany | POM02 | 2009 | - | - | - | - |




**FIGURES**

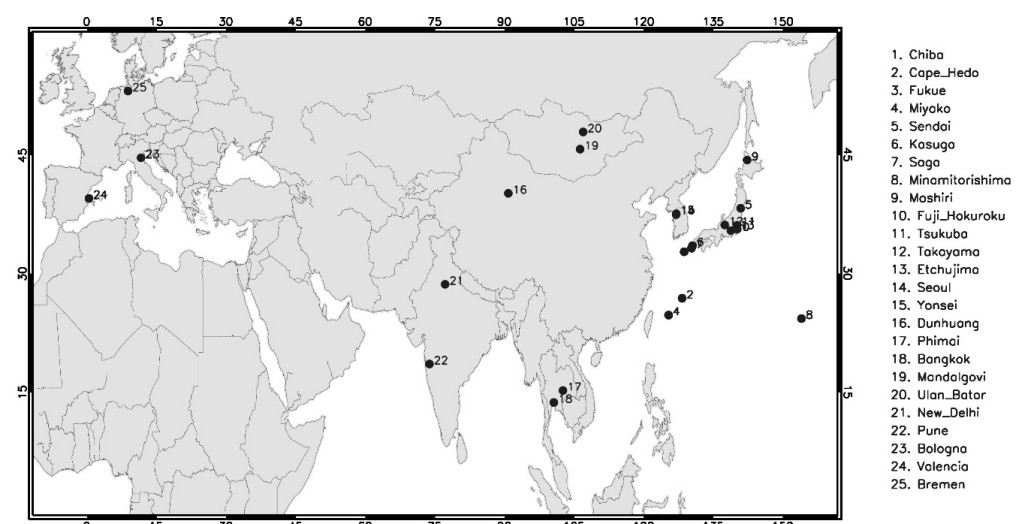

**Figure 1** Geographical placement of ground-based SKYNET sensors over sites in Asia and Europe
The SKYNET dataset for these sites are freely accessible from the Center for Environmental
Remote Sensing (CERes), Chiba University, Japan (http://atmos3.cr.chiba-
u.jp/skynet/data.html).



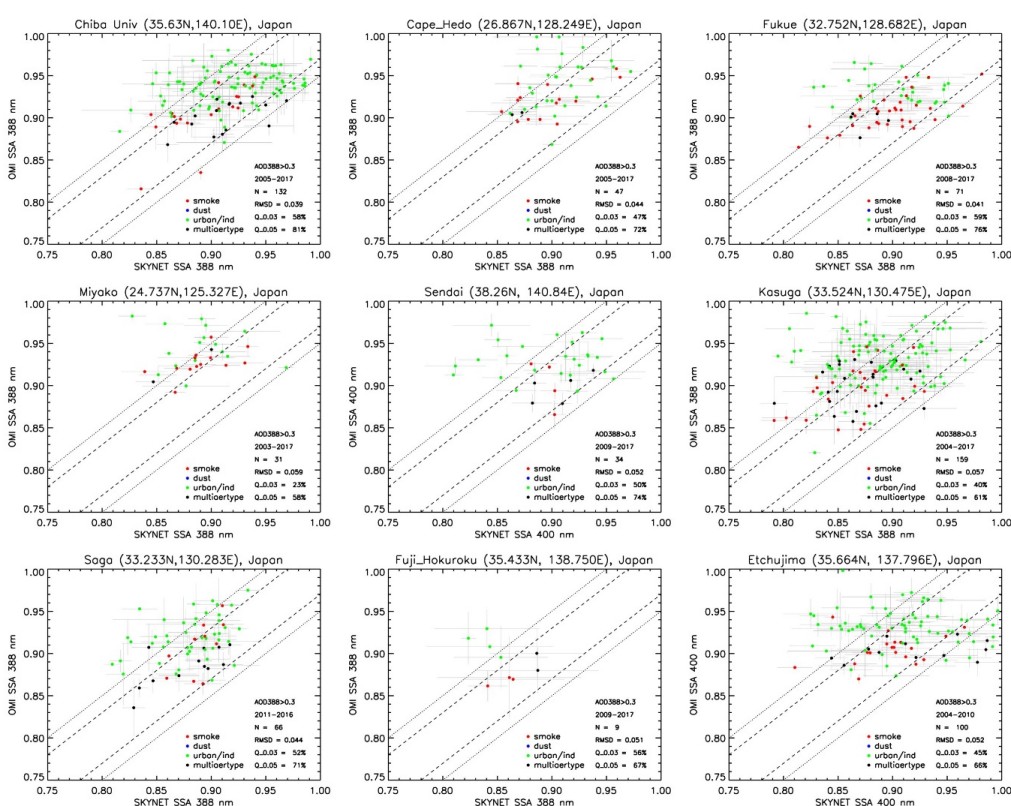


**Figure 2** OMAERUV versus SKYNET single-scattering albedo comparison for different sites in Japan. Legends with different colors represent the aerosol type selected by the OMAERUV algorithm for the co-located matchups (N). RMSD is the root-mean-square difference between the two retrievals; Q_0.03 and Q_0.05 are the percents of total matchups (N) that fall within the absolute difference of 0.03 and 0.05, respectively. OMI-SKYNET matchups with AOD>0.3 (388 or 400 nm) in both measurements are used for comparison.







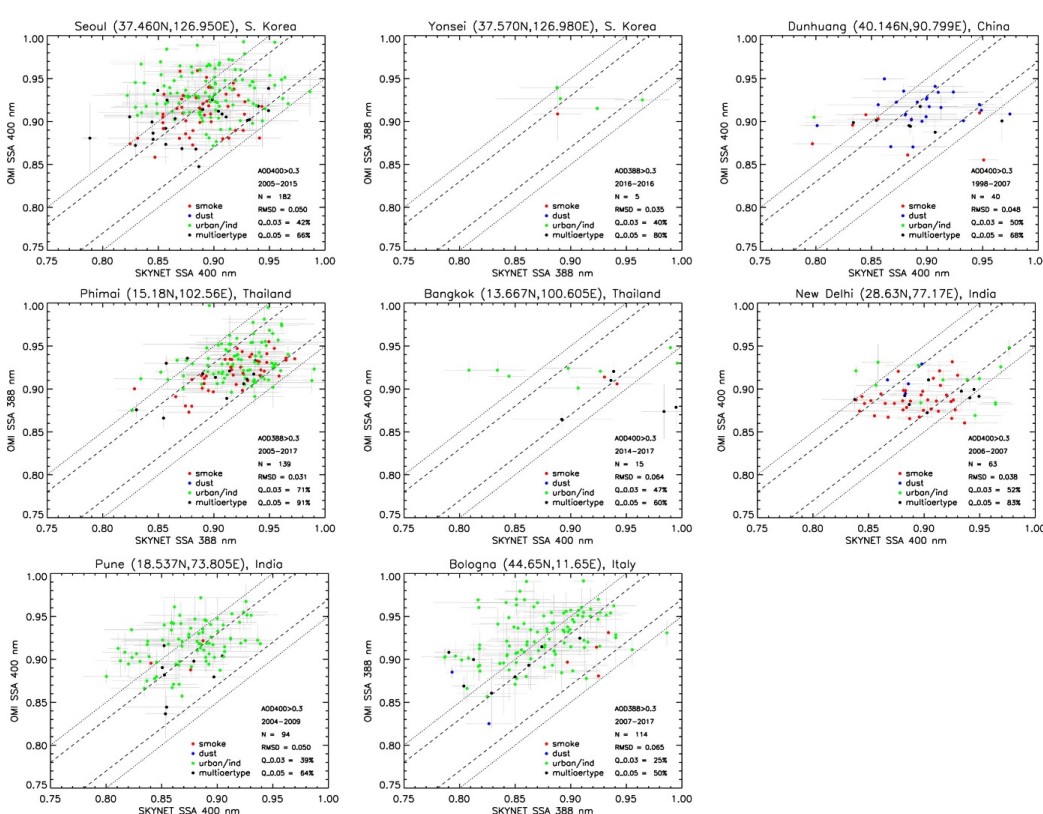


**Figure 3** Same as in Figure 2 but for SKYNET sites in South Korea, China, Thailand, India, and
Italy.

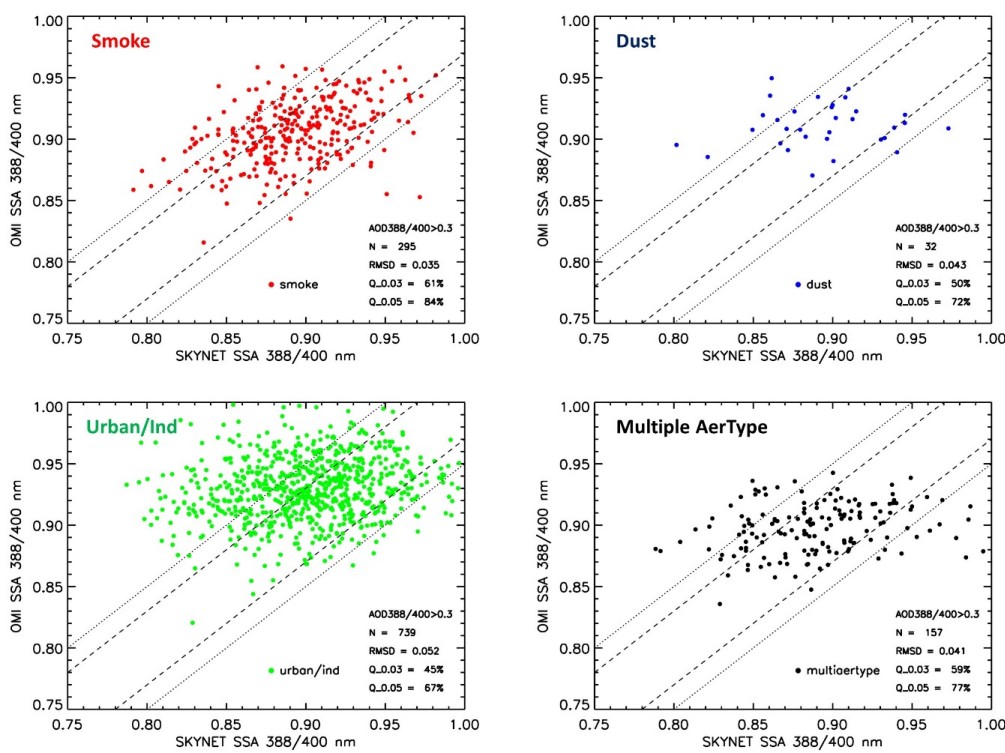


**Figure 4** Composite scatterplots of OMAERUV versus SKYNET single-scattering albedo (388 or 400 nm) for the three distinct aerosol types, i.e., smoke, dust, and urban/industrial, as identified by the OMAERUV algorithm. OMI-SKYNET matchups with AOD>0.3 (388 or 400 nm) in both measurements are used for the comparison.

736

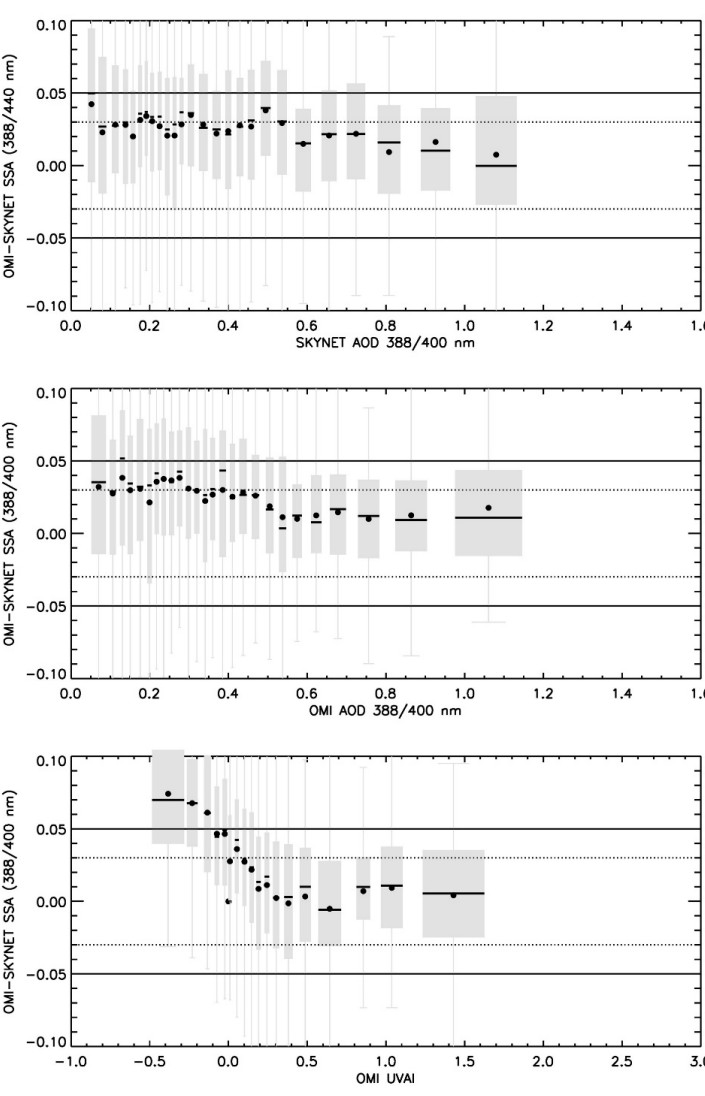

737

**Figure 5** Difference in single-scattering albedo between OMI and SKYNET as a function of the coincident SKYNET-measured (top panel) and OMI-retrieved (middle panel) aerosol optical depth and OMI-measured UVAI (bottom panel). Filled circles in black are the mean of difference for each AOD and UVAI bin with an equal sample size of 150 matchups; horizontal lines represent median of the bin samples; shaded area in gray encompasses data within 25 (lower) to 75 (higher) percentile range, whereas vertical lines in gray represent 1.5 times inter-quartile range (25 to 75 percentile). The dotted and solid horizontal lines are the uncertainty range of ±0.03 and ±0.05 respectively.





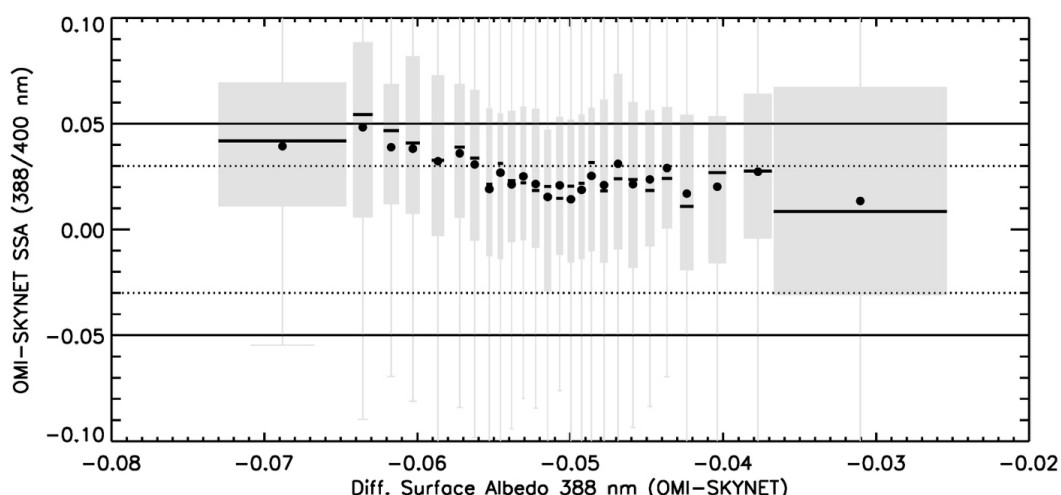

746

747 **Figure 6** Same as in Figure 5 but the difference in SSA between OMI and SKYNET is related to

748 the difference in surface albedo assumed by the two algorithms.