# Peer review of "A Comparative Evaluation of Aura-OMI and SKYNET Near-UV Single-scattering Albedo Products"

_Atmospheric Measurement Techniques, 2019_

## Referee Comment (RC2) · Anonymous Referee #2 · 16 Jul 2019

**Review report amt-2019-174**

The present study deals with the assessment of SSA derived by the OMI sensor against the corresponding retrievals obtained from ground SKYNET stations. Most of the issues addressed here have been already discussed in Jethva et al. (2014, JGR) and therefore the contribution of the current work is relatively small. An "innovative" aspect is the availability of coincident OMI-SKYNET UV wavelengths (in contrast to AERONET) thus allowing the direct intercomparison between spaceborne and ground-based measurements. Nevertheless, a significant portion of the ground retrievals is coming from POM01 instruments (longer wavelengths of 400nm) requiring the extrapolation of OMI SSA based on the assumed aerosol model in the retrieval algorithm. Below are listed my comments which should be clarified by the authors prior the publication of the submitted manuscript.

1. **Lines 225-227:** The selection of degrees (variable with latitude) instead of distance has impact on the area which is averaged for the OMI retrievals. How much important is this for the overall results?
2. **Lines 235-237:** How much reliable are the aerosol models considered in the OMI retrieval algorithm? This is critical since an inappropriate selection of an aerosol model can lead to unrealistic SSA retrievals and subsequently will have impact on the intercomparison outputs.
3. **Line 244:** Please provide an explanation about the uncertainty limits (Q_0.03 and Q_0.05). Have they been arbitrarily selected?
4. **Section 3.3:** Are there any quantified dependencies on the solar zenith angle, station altitude or aerosol layer height?
5. **Lines 389-395:** Unfortunately, the number of OMI-SKYNET pairs is limited in order to extract robust results. Moreover, how much can affect the irregularity of dust particles' shape? (see Gasteiger et al. (2011); https://www.tandfonline.com/doi/pdf/10.1111/j.1600-0889.2011.00559.x?needAccess=true
6. **Lines 431-436:** These two sentences are confusing. Since the absorption signal increases with height why the SSA values are also increase (i.e., less absorbing particles)?
7. **Lines 442-444:** How exactly MODIS is used in the OMI near-UV surface albedo database?
8. **Section 4.2:** In the discussion of this section there are not plots quantifying the OMI-SKYNET declinations with respect to cloud contamination, aerosol layer height and the prescribed surface albedo.
9. **Figures 4-5-6:** Which is the difference on the obtained overall results when POM01 and POM02 SKYNET retrievals are grouped separately?

---

## Referee Comment (RC3) · Anonymous Referee #2 · 16 Jul 2019

Please ignore the first version of the review report for the manuscript amt-2019-174. The correct report has been resubmitted.
* * *

---

## Referee Comment (RC4) · Anonymous Referee #1 · 19 Jul 2019

Review of

"A Comparative Evaluation of Aura-OMI and SKYNET Near-UV Single-scattering Albedo Products"

The paper is very interesting because it is the first time a comparison study is performed using a large number of SKYNET sites and products from this network. In addition, the results are very important for the developers of Skyrad pack improvements, particularly for what concern the assumption of fixed value of Surface Albedo.

Below some specific comments to the paper:

Lines: 65-66 specify if the change of estimated radiative forcing refers to the top, bottom or middle atmosphere.

Line 170: both the POMs models take also measurements at 315 and 940 nm for Ozone and water vapour retrieval. Add this information here and in line 178.

Line 186: remove and between University and Valencia

Line 255: add "carbonaceous/smoke"

Lines 255-257: it not clear to me according to which parameter has been considered the 5 listed sites better than the others. Moreover looking at Figure 2 the largest percentage of agreement is for Q_0.05 and not 0.03.

Line 348, it is better specify that the assumption of fixed ground albedo in Skyrad pack can be changed in time and wavelengths, if necessary. . For example in ESR/SKYNET, Antarctica sites are processed with different values.

I still suggest using greater characters for Figure 2, 3, 4. It has been difficult reading the statistics values.

---

## Author Comment (AC1) · 25 Sep 2019

The referee is encouraged to refer our response to comments and concerns raised by the anonymous referee # 2, where we have incorporated additional analysis to the manuscript.

RC: The paper is very interesting because it is the first time a comparison study is performed using a large number of SKYNET sites and products from this network. In addition, the results are very important for the developers of Skyrad pack improvements, particularly for what concern the assumption of fixed value of Surface Albedo. AR: We appreciate the reviewer for the constructive comments. In the OMI-SKYNET compari-

son, we have considered all SKYNET stations whose data are freely accessible online from Chiba University SKYNET server. As the referee as stated here, the purpose of such comparison, particularly when both quantities are not directly measured but retrieved using respective algorithms, is to understand and diagnose the (dis)agreement between the two datasets to improve the accuracy of both retrievals.

RC: Lines: 65-66: specify if the change of estimated radiative forcing refers to the top, bottom or middle atmosphere. AR: The sentence is modified as "Together, both AOD and SSA determine the magnitude and sign of the aerosol radiative forcing at the top-of-atmosphere."

RC: Line 170: both the POMs models take also measurements at 315 and 940 nm for Ozone and water vapour retrieval. Add this information here and in line 178. AR: The header information given in both POM-01 and POM-02 datasets states that the former sensor carries a total of five wavelength filters covering visible to near-IR (400-1020 nm), whereas the latter has two additional filters in the UV region (340 and 380 nm) along with the other filters in the visible to shortwave-IR (including 1627 nm and 2200 nm) part of the spectrum. These data files do not mention the use of 315 and 940 nm for Ozone and water vapor retrievals.

RC: Line 186: remove and between University and Valencia AR: Corrected.

RC: Line 255: add "carbonaceous/smoke" AR: Corrected.

RC: Lines 255-257: it not clear to me according to which parameter has been considered the 5 listed sites better than the others. Moreover looking at Figure 2 the largest percentage of agreement is for Q_0.05 and not 0.03. AR: The sentence referred here is for the carbonaceous/smoke aerosol type (red dots in the scatterplots), for which the majority of matchups are confined within the difference of 0.03. For the overall compassion between the sites, we considered RMSD and % matchups falling within 0.03/0.05 as criteria. Lower RSMD and higher % matchups (Q_0.03 & Q_0.05) suggest a better comparison between OMI and SKYNET SSAs.

RC: Line 348, it is better specify that the assumption of fixed ground albedo in Skyrad pack can be changed in time and wavelengths, if necessary. For example in ESR/SKYNET, Antarctica sites are processed with different values. AR: The first two sentences are revised as "The standard SKYNET inversion algorithm assumes a wavelength-independent surface albedo of 0.1 at all wavelengths across the UV to visible part of the spectrum. However, the algorithm code allows flexibility to alter the value surface albedo in time and wavelength (Campanelli et al., 2015)."

RC: I still suggest using greater characters for Figure 2, 3, 4. It has been difficult reading the statistics values. AR: Figure 2, 3, and 4 are reproduced with bigger size characters and numbers.

---

## Author Comment (AC2) · 25 Sep 2019

RC: Referee's comment AR: Author's response

RC: The present study deals with the assessment of SSA derived by the OMI sensor against the corresponding retrievals obtained from ground SKYNET stations. Most of the issues addressed here have been already discussed in Jethva et al. (2014, JGR) and therefore the contribution of the current work is relatively small. An "innovative" aspect is the availability of coincident OMI-SKYNET UV wavelengths (in contrast to AERONET) thus allowing the direct intercomparison between spaceborne and ground-based measurements. Nevertheless, a significant portion of the ground retrievals is

coming from POM01 instruments (longer wavelengths of 400nm) requiring the extrapolation of OMI SSA based on the assumed aerosol model in the retrieval algorithm. Below are listed my comments which should be clarified by the authors prior the publication of the submitted manuscript.

AR: While we agree with the reviewer that the present study follows our earlier paper of OMI-AERONET SSA comparison (Jethva et al., 2014), the present study is the first-ever attempt to evaluate the OMI SSA retrievals against those of SKYNET. The POM-02 sensors owing to its retrievals in the near-UV region facilitates direct comparison with those of OMI at 388 nm. The POM-01 sensors, on the other hand, do not carry UV wavelength filters, but offer retrievals at 400 nm requiring an extrapolation of OMI SSA to only 12-nm window against 52-nm when compared with AERONET at 440 nm. It is reasonably fair to assume that the extrapolation of OMI SSA in a narrow window of only 12-nm should not be a major source of uncertainty in comparing SSA from OMI and SKYNET. An additional analysis comparing OMI SSA with POM01 and POM02 dataset separately, which is discussed later in this response, suggests that the resultant OMI-SKYNET statistics do not change significantly regardless of the use of POM01 or POM02 in the evaluation.

In the revised manuscript, it is stated that,

"It is reasonably fair to assume that the extrapolation of OMI SSA in a narrow window of 12-nm, i.e., from 388 to 400 nm, shouldn't be a major source of uncertainty in comparing SSA from OMI and SKYNET."

RC: 1. Lines 225-227: The selection of degrees (variable with latitude) instead of distance has impact on the area which is averaged for the OMI retrievals. How much important is this for the overall results? AR: We attempted several spatial windows for the OMI averaging, i.e., $0.1°$, $0.25°$, $0.5°$, or even $1.0°$. However, the statistical results in terms of RSMD and % matchups within 0.03/0.05 and derived conclusion didn't alter much. We choose a spatial window of $0.5°$ centered at SKYNET station to

be consistent with our earlier work of comparing OMI and AERONET SSA.

RC: 2. Lines 235-237: How much reliable are the aerosol models considered in the OMI retrieval algorithm? This is critical since an inappropriate selection of an aerosol model can lead to unrealistic SSA retrievals and subsequently will have impact on the intercomparison outputs.

AR: The aerosol size distribution and spectral dependence of AOD assumed for all three standard aerosol types considered in the OMAERUV algorithm are based on the AERONET ground-based inversions reported in Dubovik et al. (2002). The spectral dependence of absorption, often quantified as Absorption Angstrom Exponent, for the carbonaceous smoke aerosols accounts for the presence of organics in biomass burning aerosols and referenced to the in-situ absorption measurements taken during SAFARI-2000 experiment (Kirchstetter et al., 2004; Jethva and Torres, 2011). In the latest OMAERUV product used in the present study, the urban-industrial aerosol model is also revised to adopt the same spectral dependence of absorption that of the smoke model. The expected uncertainty of ±0.03 in the retrieved-SSA accounts for an error in the selection of aerosol model.

RC: Line 244: Please provide an explanation about the uncertainty limits (Q_0.03 and Q_0.05). Have they been arbitrarily selected? AR: The uncertainty limits of OMAERUV SSA retrievals, i.e., ±0.03 and ±0.05 were determined based on its comparison with AERONET SSA inversion reported in Torres et al. (2007). Following the early evaluation of OMI aerosol product for a handful of sites, Jethva et al. (2014) conducted a global evaluation of SSA product and also conducted a detailed uncertainty test considering different sources of errors, such as aerosol model, surface albedo, and aerosol layer height. The results of the sensitivity analysis suggested that overall, OMI SSA retrievals are uncertain to within 0.03 to 0.05, however errors could be even larger when algorithmic assumptions are far off from the real atmospheric conditions. This has been clarified in the section 2.1 of the revised paper.
RC: Section 3.3: Are there any quantified dependencies on the solar zenith angle, station altitude or aerosol layer height? AR: The OMAERUV algorithm does consider the variation in altitude over land in the inversion by interpolating the radiances in terrain pressure in logarithmic space. Therefore, we don't expect significant error in the OMI SSA retrievals due to varying terrain pressure from one station to another. The aerosol layer height (ALH) assumed in each valid retrieval is referenced to the 30-month long CALIOP-OMI 1° x 1° global database [Torres et al., 2013]. Results obtained from a sensitivity study reported in Jethva et al. [2014] suggested that a decrease in the assumed ALH from 3 km to 1.5 km for the dust aerosol type results in a decrease in SSA by 0.02. Although, the climatology dataset of CALIOP-OMI adequately describes the observed mean layer of carbonaceous and desert dust aerosols, variations in the ALH not observed by CALIOP, therefore, will be missed out in the derived climatology and thus can be a source of uncertainty. Assuming that the CALIOP-OMI ALH dataset is accurate to within ±1 km, the resultant error in SSA retrieval would be ∼ ±0.01.

The dependence of SSA difference on the local hour of SKYNET measurements are quantified in the Figure shown below. The SKYNET dataset accessed from the data server at Chiba University doesn't contain information on the solar zenith angle. However, the local time of measurements reported in data file for each station can serve a proxy for the solar zenith angle. The OMI-SKYNET matchups exhibit a systematic dependency, where the differences between the two datasets become relatively minimal when early morning and late afternoon inversions of SKYNET associated with higher solar zenith angle are collocated with OMI retrievals around 1:30 PM local time overpass. Owing to a longer atmospheric optical path at higher solar zenith angle, thereby better aerosol signal, the ground-based aerosol inversions, such as from AERONET and SKY NET, are expected to be more reliable for sky measurements carried out during early morning/late afternoon.

Figure 1 The difference in SSA between OMI and SKYNET is related to the surface albedo difference and local measurement hour of SKYNET.

The results obtained in this analysis are added to the discussion in the revised manuscript.

RC: Lines 389-395: Unfortunately, the number of OMI-SKYNET pairs is limited in order to extract robust results. Moreover, how much can affect the irregularity of dust particles' shape? (see Gasteiger et al. (2011); https://www.tandfonline.com/doi/pdf/10.1111/j.1600-0889.2011.00559.x?needAccess=true

AR: The collocation procedure yields fewer OMI-SKYNET matchups for the dust aerosol type identified with the OMAERUV algorithm. The sensitivity study followed by an actual inversion of OMI data presented in Torres et al. (2018) demonstrates that the change in dust particle shape from spherical to spheroidal distribution improved the AOD retrievals significantly and brought the equivalency between the retrievals over left and right sides of the OMI swath for the dust belt region of tropical Atlantic. The associated changes in SSA retrievals were noted within plus/minus 0.01 and -0.02 for the scattering angle up to 100-150 degree and >160degree, respectively. The OMAERUV version 1.8.9.1 data product used in the present study adopts spheroidal dust model based on the work of Dubovik et al. (2006) and Torres et al. (2018). We have elaborated the discussion on this aspect of the OMI aerosol retrievals in the uncertainty section 4.2.

RC: Lines 431-436: These two sentences are confusing. Since the absorption signal increases with height why the SSA values are also increase (i.e., less absorbing particles)? AR: We realized that the information presented here is miscommunicated. The corrected sentence should be,

"This is because an increase (decrease) in the assumed aerosol layer height from the actual one enhances (reduces) absorption in the radiance look-up table (not in the actual TOA measurements), which the OMAERUV algorithm compensates by retrieving lower (higher) AOD and higher (lower) SSA to match with the observations."

RC: Lines 442-444: How exactly MODIS is used in the OMI near-UV surface albedo database? AR: The new OMI-based near-UV surface albedo database was essentially derived based on the minimum reflectivity approach. The 7-year record of OMI observation of Lambertian Equivalent Reflectivity or LER at 388 was used to find its minimum value corresponding to UVAI<0.5 for each grid (0.25ïĆřx0.25ïĆř) and for each month. The 354-nm LER values are anchored to the corresponding minimum LER at 388 nm. During the persistent cloudy period for grids identified with higher LER values and abrupt temporal variations, the information on monthly variations of high-res MODIS surface albedo dataset aligned with the corresponding temporal record of OMI minimum LER was used to estimate LER in the near-UV.

RC: Section 4.2: In the discussion of this section there are not plots quantifying the OMI-SKYNET declinations with respect to cloud contamination, aerosol layer height and the prescribed surface albedo. AR: The dependence of the SSA difference between OMI and SKYNET as a function of difference in the surface albedo assumption are already shown in Figure 6 and discussed in the section 4.2 in the manuscript. Regarding the cloud contamination, we use the 'best' quality retrievals from both sensors OMI (final algorithm quality flag=0) and SKYNET (cloud flag=0) in this study largely eliminating the chances of significant cloud interference in both kinds of measurements. Given the larger footprint size of OMI (13x24 km2), however, the possibility of sub-pixel cloud contamination cannot be ruled out completely. On the other hand, a good level of agreement between the two retrievals at higher aerosol loading, and when the differences in the surface albedo reduces to minimum, are indicative of a consistency under favorable conditions. The explanation on the aerosol layer height assumption is explained in response to one the previous comments.

RC: Figures 4-5-6: Which is the difference on the obtained overall results when POM01 and POM02 SKYNET retrievals are grouped separately? AR: Figure displayed below shows the number density plot comparing SSA between SKYNET and OMI for POM01 and POM02 sensors separately. Overall, we don't see a major difference in the derived

statistics between the two sets of comparison, except that the number of matchups obtained with POM02 sensor is 39% more than those with POM01 sensors and POM02 dataset offers marginally better comparison (except bias, which is higher with POM02) with OMI SSA retrievals. This analysis indicates that the interpolation of OMI SSA from 388 nm to 400 nm for its comparison with POM01 data isn't a significant source of discrepancy between the two SSA datasets. The revised manuscript includes the new analysis and related discussion.

Figure 2 Number density plots of SSA comparison between OMI and SKYNET matchups derived separately using POM-01 (left) and POM-02 (right) sensors. The resultant statistics of the comparison are depicted in the lower-right in each plot.
* * *
[Figure]

[Figure]

**Fig. 1.**

**POM01 SKYNET Sites**

| Sendai | Etchujima | Seoul | Dunhuang | Bangkok |
| Mandalgovi | Ulanbator | Pune | Valencia | |

**POM02 SKYNET Sites**

| Chiba | Hedo | Fukue | Miyako | Kasuga |
| Saga | Minamitorishima | Moshiri | Fujihokuroku | Tsukuba |
| Takayama | Yonsei | Phimai | Bologna | Bremen |
| Lauder | | | | |

N = 1123
RMSD = 0.061
Bias = 0.021
Q_0.03= 36.42%
Q_0.05= 57.35%

N = 1568
RMSD = 0.059
Bias = 0.028
Q_0.03= 38.97%
Q_0.05= 60.71%

Number Density of OMI−SKYNET Matchups

Number Density of OMI−SKYNET Matchups

**Fig. 2.**

---

## Author Comment (AC3) · 27 Sep 2019

Comments and suggestions made by referee # 2 on amt-2019-174 are responded as AC2 following the corrected version of the referee's report RC2.
* * *